# Using insurance data to quantify the multidimensional impacts of warming temperatures on yield risk

Edward D. Perry[1✉], Jisang Yu [1] & Jesse Tack[1]

Previous research predicts significant negative yield impacts from warming temperatures, but estimating the effects on yield risk and disentangling the relative causes of these losses remains challenging. Here we present new evidence on these issues by leveraging a unique publicly available dataset consisting of roughly 30,000 county-by-year observations on insurance-based measures of yield risk from 1989–2014 for U.S. corn and soybeans. Our results suggest that yield risk will increase in response to warmer temperatures, with a 1 °C increase associated with yield risk increases of approximately 32% and 11% for corn and soybeans, respectively. Using cause of loss information, we also find that additional losses under warming temperatures primarily result from additional reported occurrences of drought, with reported losses due to heat stress playing a smaller role. An implication of our findings is that the cost of purchasing crop insurance will increase for producers as a result of warming temperatures.

[1] Department of Agricultural Economics, Kansas State University, 342 Waters Hall, 1603 Old Claflin Place, Manhattan, KS 66506, USA. ✉email: edperry@ksu.edu

Effective adaptation of crop production to climate change has widespread implications for global food security given projected increases in world population[1–5]. It is well established that crop yields exhibit sensitivity to climate change through the effects of temperature, precipitation, and atmospheric $CO_2$ concentration[6–15]. Warmer temperatures are often associated with large, negative effects on crop yields at regional and global scales through both direct (e.g., heat stress) and indirect (e.g., soil moisture deficit) mechanisms[16]; however, it is often difficult to disentangle these effects from one another using large-scale observation-based datasets[10,17–24]. This is problematic because the ability of production agriculture to reduce climate change impacts depends crucially on this disentanglement for both technological innovation (e.g., targeted breeding efforts) and on-farm decision-making (e.g., crop choice, cultivar selection, and planting dates)[25–32].

Most previous work linking climate to crop yields has focused on changes to average or *expected* yields, which is useful information for future food security concerns as supply chains and consumers can begin to readjust expectations. However, there still exists the possibility of shortfalls below these new expected yields, and these shortfalls are often associated with large price spikes and social unrest[33–40]. Furthermore, large unexpected shocks to farm revenue can reduce on-farm production investments and even bankrupt farmers, thereby causing disruptions in food supply[41,42]. In addition, knowledge on the propensity for below average food production and revenue outcomes has implications for the large and rapidly expanding global crop insurance sector, where large indemnity payments can lead to high costs of operating public crop insurance programs[43–47]. In recognition of this, recent climate change studies have begun focusing on the variability of farm production[10,43,48–51].

In this paper, we utilize a novel and large-scale dataset that provides insights into three major dimensions of warming temperature effects: the propensity of both yield and revenue outcomes below expected levels, the disentanglement of heat-stress versus drought-stress on these outcomes, and the pricing of crop insurance. To conduct the empirical analysis, we rely on a relatively untapped data source on crop insurance indemnity payments: the Cause of Loss (COL) database[52], which is publicly available and maintained by the Risk Management Agency (RMA). We focus on U.S. corn and soybean production using annual data from 1989 to 2014 containing 30,261 (29,014) total observations over 1733 (1632) counties for corn (soybeans). To the best of our knowledge, very few studies have used the COL data[53–55], despite its ability to provide insights into both the magnitude of losses and their causes.

## Results

### COL data and the loss–cost ratio (LCR).
The COL data provide separate measures of yield and revenue losses below expected levels (paid indemnities). In addition, the COL data are disaggregated by type of cause, which is determined by a claims adjuster upon determination of payment; in total, we observe 44 reported causes. A particularly useful aspect of the COL data is that it distinguishes between moisture-related (e.g., drought and excess moisture) and purely temperature-related causes (e.g., heat stress) (Fig. 1). We merge the COL database with the Summary of Business (SOB) database[56], a separate RMA database that contains county-level aggregate liabilities by crop. The merged data is then used to generate county-level aggregate and cause-specific variables equal to the ratio of losses to liabilities, or the LCR: $LCR_{it} = \left(\sum_j Losses_{it}^j\right)\left(Liabilities_{it}\right)^{-1}$, where $Losses_{it}^j$ is the dollar value of realized indemnity payments in county $i$ in year $t$ due to cause $j$ and $Liabilities_{it}$ is the maximum dollar value of indemnity payments in

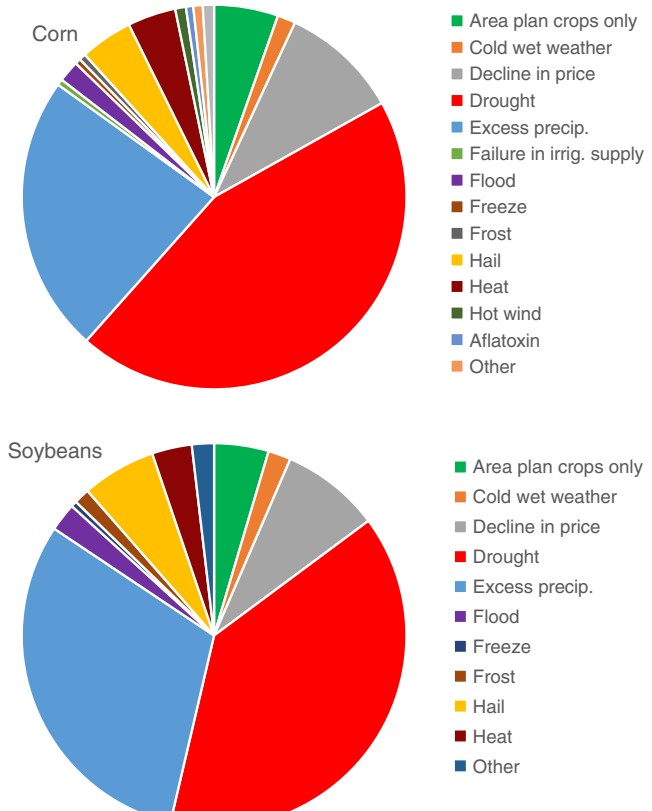

**Fig. 1 Indemnity payment shares by cause of loss, 1989–2014.** Each portion of the pie chart represents the average share of indemnity payments in each crop due to a particular cause.

county $i$ in year $t$. A key advantage of using the LCR variable instead of mean yields (as used in Schlenker and Roberts[6], among others) is that it constitutes a measure of yield risk. While previous studies find that mean yields are predicted to decrease as a result of warming, these studies do not capture whether production will become more or less risky under warming. Indeed, the empirical finding on the negative impact of warming on mean yields in the literature does not necessitate that warming also increases downside yield risk around the new mean yields; these directional impacts need to be measured separately. Moreover, an appealing feature of the county-level LCR or crop insurance loss measures is that they capture downside risks of individual farms, whereas the county-level average yield data may not vary in response to idiosyncratic losses by farmers within a county. As a result, analyses that rely on yield data only will tend to underestimate the impact of warming on yield sensitivity. For a simulation-based demonstration of this, see Supplementary Discussion and Supplementary Table 6.

The county–year LCR data are matched to growing-season aggregates of precipitation and temperature exposure following the panel approach of Schlenker and Roberts[6]. Multivariate regression analysis is used to estimate the relationship between LCRs and weather covariates while controlling for county-level time-invariant confounders (county fixed effects), year fixed effects, and state-specific time trends. The LCRs, temperature, and precipitation vary substantially across both counties and years (Supplementary Fig. 1). Following previous studies, we focus on U.S. dryland counties, i.e., counties to the east of the 100th degree meridian. Some of these counties may rely partially on irrigation, but most corn grown in this region is rain-fed. Within this region, time-averaged LCRs demonstrate significant cross-sectional variation (Fig. 2).

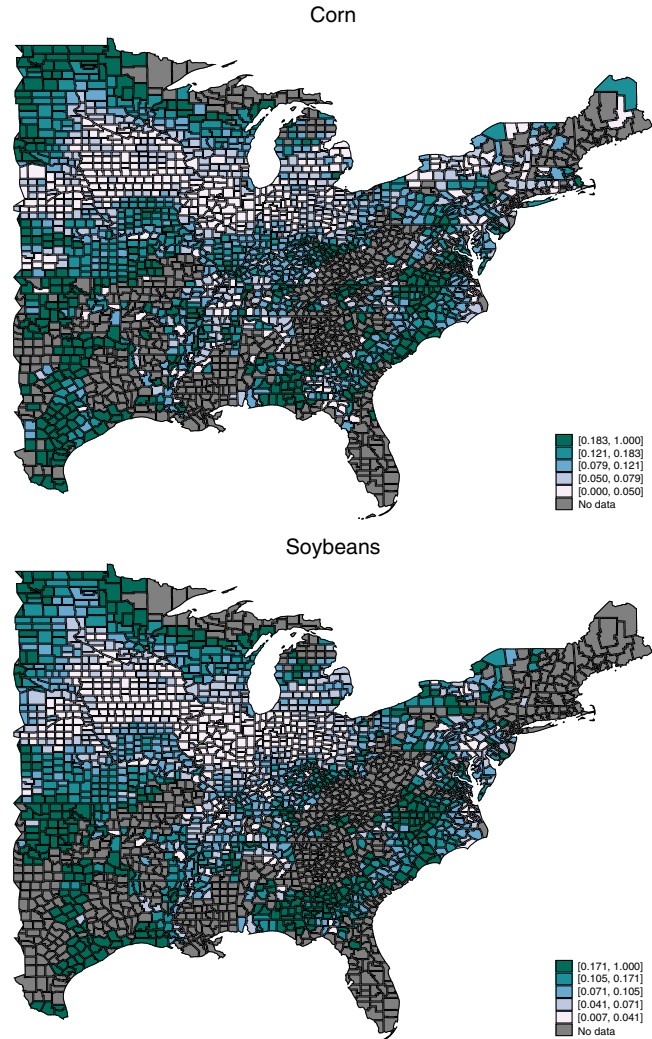

**Fig. 2 Spatial distribution of county-level aggregate loss–cost ratios (LCR), 1989–2014 averages (N = 30,261 for corn and N = 29,014 for soybeans).** The darker colors represent higher average LCRs.

We focus on five types of LCRs. The first type we consider is the total or aggregate LCR, which includes losses across all causes. The remaining four types are cause specific: heat, cold, drought, and excess moisture (The *cold* losses are the sum of cold wet weather, freeze, and frost losses). Drought exhibits the highest LCR of the four, is associated with approximately 50% of total losses, and is double the next largest cause (excess moisture), on average (Table 1). The average LCRs for heat and cold are similar in magnitude and roughly an order of magnitude lower than drought. It is important to recognize, however, that the determination of what causes a particular loss is based on the interpretations of adjusters and producers, which may not always align exactly with the true cause of losses. Nonetheless, the perceptions of adjusters and producers are just as important for understanding the ramifications of climate change, as they likely reflect how farmers adjust their behavior in response to warmer temperatures.

Annual variation between high temperatures and total LCRs at the aggregate level (across counties) exhibit positive correlation (Supplementary Fig. 2). Losses attributed to temperature extremes, excessive heat and cold, have varied over time but with opposite trends: heat losses are becoming more common while cold losses are becoming less so (Supplementary Fig. 3).

Moisture-related losses, drought and excess moisture, appear to be more stable with no clear trends emerging over time (Supplementary Fig. 4). Degree days over 29 °C are positively correlated with the total, heat, and drought LCRs, which is consistent with high temperatures having both direct and indirect effects on yields (Table 1). Based on within-state variation, we also observe positive correlations between total, heat, and drought LCRs and degree days above 29 °C (Supplementary Fig. 5). Precipitation is less correlated with LCRs (Table 1) in total and for all causes except excess moisture; however, simple correlations likely understate the relationship given its nonlinear effect on yields.

**Estimated weather impacts using multivariate regression**. The multivariate regression model permits a broader understanding of the relationships between weather and losses. We first regress total LCRs at the county–year level against a piecewise linear function of temperature exposure and a quadratic function of cumulative precipitation. To control for unobserved sources of variation in the LCR, we included county fixed effects, year fixed effects, and state-specific time trends. County fixed effects control for unobserved factors, such as soil quality. The year fixed effects and state-specific trends control for time-varying unobserved factors, such as changes in the federal crop insurance program, as well as changes in behavior (e.g., enrollment, coverage level, contract choice). Goodness-of-fit and temperature cutoff results are reported in Supplementary Table 1. Further details on the regression model can be found in the "Methods" section.

The marginal effects of the weather variables exhibit highly nonlinear relationships between temperatures and precipitation with losses (the marginal effects of temperature and precipitation are reported in Fig. 3 and Supplementary Figs. 6 and 7). In the top row of Fig. 3, the slopes of the piecewise linear function correspond to the estimated temperature coefficients from Eq. (1). These graphs depict the impact of one additional day at each temperature on the overall LCR. For example, one additional day at 27 °C instead of a day below 10 °C within the fixed growing season reduces the LCR by a bit under 0.01 or one percentage point (or, put differently, by one cent per dollar of liability). Conversely, one additional day at 35 °C raises the LCR by just over 0.01. In corn, warmer temperatures begin to raise the LCR at about 30 °C and these effects intensify up to 40 °C. Similar estimated impacts are depicted for soybeans. We also note that the estimated temperature impacts are very similar to the estimated impacts in Schlenker and Roberts[6]: they find sharp decreases in *yields* beginning at 29 °C for corn and at 30 °C for soybeans. We also observe that additional exposure to so-called beneficial heat—temperatures below the first threshold—reduce the LCRs and at an increasing rate. Concerning precipitation, the estimated impacts reflect a concave relationship between crop yields and precipitation: at lower precipitation levels, additional rainfall reduces the LCR, but at higher levels of precipitation, it raises the LCR (Supplementary Fig. 7).

The middle and lower panels of Fig. 3 highlight the role of temperature on heat and drought losses. In corn, warmer temperatures are associated with increased heat stress indemnities beginning at 30 °C, with the effects intensifying at 38 °C. Drought losses begin to increase significantly at 29 °C. In soybeans, heat and drought losses follow similar patterns, albeit at slightly higher thresholds: heat losses begin to increase at 33 °C and drought losses begin to increase at 30 °C. The estimated precipitation impacts follow different patterns (Supplementary Fig. 7). For both corn and soybeans, at lower levels of precipitation, additional rainfall reduces the losses associated with drought; however, at extreme levels those effects diminish to zero and even become

**Table 1 Summary statistics for LCR, temperature, and precipitation variables, 1989–2014.**

| Cause | Average | Correlation coefficients | | | | | | |
|---|---|---|---|---|---|---|---|---|
| | | Total LCR | Drought | Excess moisture | Heat | Cold related | Degree days 29 °C | Precipitation (mm) |
| Corn | | | | | | | | |
| Total LCR | 0.119 | 1.000 | 0.799 | 0.379 | 0.378 | 0.280 | 0.367 | −0.148 |
| Drought | 0.0619 | | 1.000 | −0.0988 | 0.309 | −0.0361 | 0.436 | −0.277 |
| Excess moisture | 0.0298 | | | 1.000 | −0.0490 | 0.115 | −0.0626 | 0.215 |
| Heat | 0.00564 | | | | 1.000 | −0.0200 | 0.244 | −0.0949 |
| Cold related | 0.00653 | | | | | 1.000 | −0.0948 | −0.0517 |
| Degree days 29 °C | 56.526 | | | | | | 1.000 | −0.186 |
| Precipitation (mm) | 612.656 | | | | | | | 1.000 |
| Soybeans | | | | | | | | |
| Total LCR | 0.107 | 1.000 | 0.782 | 0.442 | 0.292 | 0.207 | 0.408 | −0.114 |
| Drought | 0.0562 | | 1.000 | −0.0833 | 0.187 | −0.0290 | 0.467 | −0.271 |
| Excess moisture | 0.0291 | | | 1.000 | −0.0262 | 0.0498 | 0.00530 | 0.243 |
| Heat | 0.00437 | | | | 1.000 | −0.0176 | 0.314 | −0.109 |
| Cold related | 0.00396 | | | | | 1.000 | −0.0966 | −0.0447 |
| Degree days 29 °C | 54.866 | | | | | | 1.000 | −0.113 |
| Precipitation (mm) | 616.811 | | | | | | | 1.000 |

Notes: Values are based on $N = 1733$ counties for corn and $N = 1632$ counties for soybeans. Cold-related losses are the sum of freeze, frost, and cold wet weather losses.

positive in soybeans. Moreover, a 10% increase in precipitation from its mean reduces the drought LCR in both corn and soybeans by 0.008 and 0.012, respectively. Precipitation has no statistically significant effect on heat losses in soybeans or in corn.

In both crops, warmer growing seasons are associated with a reduction in both cold and excess moisture losses but these impacts are estimated with less precision (Supplementary Fig. 6). The reduction in cold losses is consistent with expectations, but the reduction in excess precipitation losses is less obvious and perhaps suggests an evaporative effect of warm temperatures. Indeed, if excessive rainfall is accompanied by high temperatures, this may reduce damages both by more quickly alleviating excess moisture conditions and by reducing the chances of damages arising from successive precipitation events in a short time span. The impacts of precipitation also generally conform with expectations with a 10% increase in precipitation from its mean increasing the excess moisture LCR in both corn and soybeans by about 0.008 and 0.006, respectively (Supplementary Fig. 7). Precipitation has no statistically significant effects on cold-related losses. The estimated impacts for soybeans largely demonstrate the same impacts as in corn.

**Impact of uniform 1 °C warming on the overall LCR.** Overall, the foregoing suggests that higher temperatures induce more heat and drought losses but reduce excess moisture- and cold-related losses. The net impacts of higher temperatures, however, cannot be easily ascertained, as contrasted with the impact of changes in precipitation, for example (reported in Supplementary Fig. 7). Therefore, we simulate a warming climate by increasing the daily minimum and maximum temperatures by 1 °C using the estimated coefficients from our five regression models (total LCR and the four cause-specific LCRs).

A uniform 1 °C warming for corn is associated with a wide range of heterogeneous impacts on the total LCR across counties ranging from −0.044 to 0.28 for corn and from −0.039 to 0.24 for soybeans (Fig. 4 and Supplementary Table 1). Generally, southern counties are adversely impacted by warmer temperatures, whereas northern counties are positively affected by warmer temperatures. For corn, the average impact across counties is 0.038 (p value = 0.023), which is approximately a 30% increase compared to the historical average of 0.12. Warming impacts on soybeans exhibit a similar pattern of results, although smaller in

magnitude and statistically insignificant (an average increase of 0.011 with a p value of 0.23 compared to a historical average of 0.11).

**Robustness of warming impacts.** To assess the robustness of the results, we estimate the models and warming impacts with different specifications regarding quantifying temperature variables (Supplementary Table 2) and the LCR computed from different subsets of insurance products (Supplementary Table 3). The warming impacts remain robust across these two alternative models. We also consider additional control variables (Supplementary Table 4) such as variables that describe crop insurance participation patterns or additional weather variables (e.g., vapor pressure deficit) and test whether adding these variables improves out-of-sample predictions. The results suggest that these alternatives do not improve model performance, and we find that the results were robust to a range of other checks. Further details on the robustness of the results can be found in Supplementary Discussion.

Previous studies highlight the importance of considering adaptation in estimating warming impacts[1,36]. In our context, in addition to adaptation in crop production, the changes in insured liabilities as responses to changes in yield history and changes in the Federal Crop Insurance Program (FCIP) also need to be considered. Thus we conduct an additional set of robustness checks with respect to the consideration of potential adaptation using an approach suggested by Hsiang[36]. We compute warming impacts based on four alternative empirical models: a long-difference model, a cross-section model, and models with 5- and 10-year moving averages of the model variables (Supplementary Table 5). As noted in Hsiang[36], estimated impacts based on approaches that exploit climatic variations over longer temporal frequencies implicitly allow for changes in beliefs or adaptation by individuals (in this case, farmers and the FCIP). Across all specifications, the impacts range from +17% (MA10) to +32% (BaseLine) for corn and from +11% (MA10) to +37% (LongDiff) for soybeans. We interpret this as strong evidence that, even with considering possible adaptations in crop production and insured liabilities, warmer temperatures tend to increase production risk and premium rates. See Supplementary Discussion for further details and discussion.

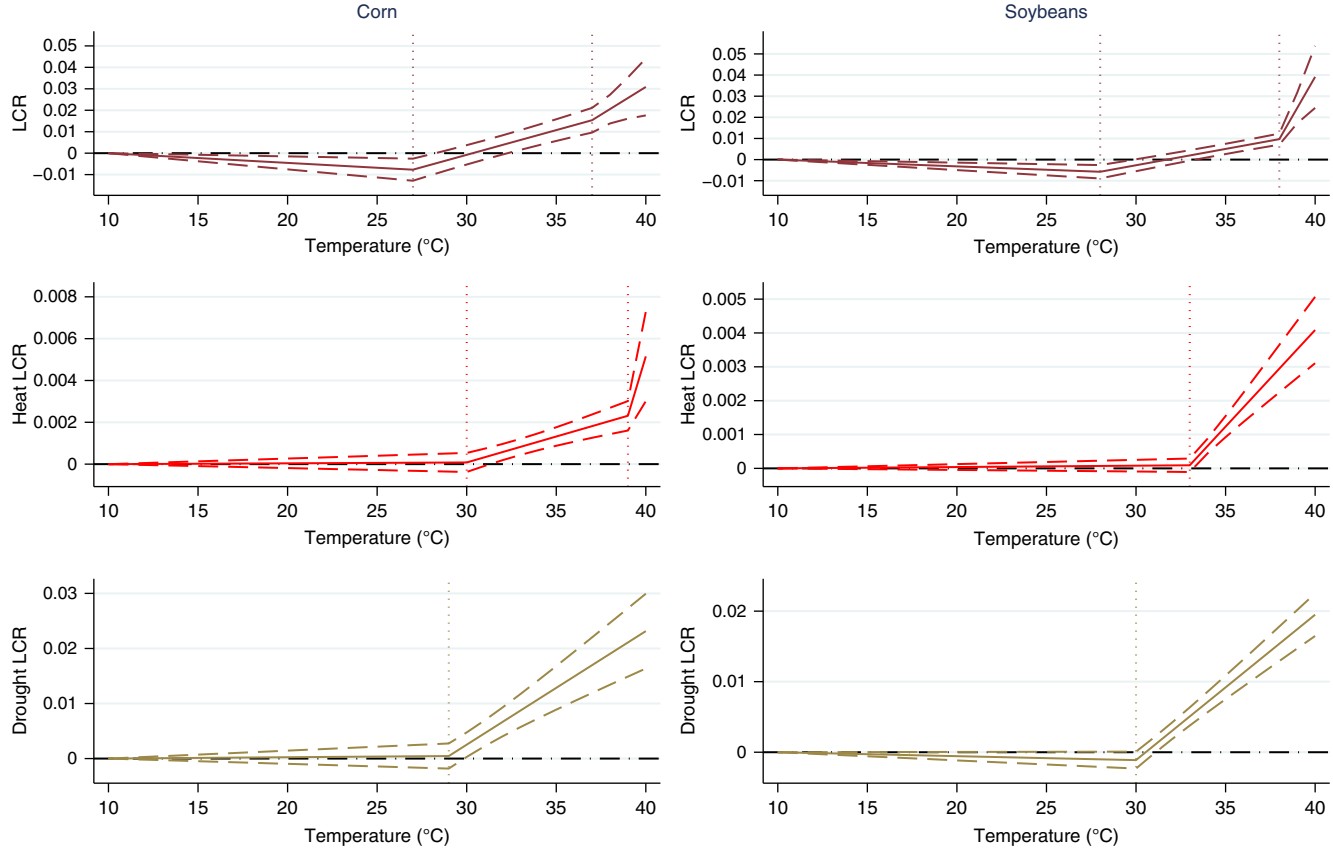

**Fig. 3 Marginal effects of temperature on the heat and drought LCRs ($N = 30,261$ for corn and $N = 29,014$ for soybeans).** Dashed lines represent 95% confidence intervals and standard errors are clustered by year. For the drought LCRs and soybean heat LCR, the second cutoffs exceed 40 °C and the marginal effects in the last temperature intervals are too large to be reported.

**Cause-specific LCR warming impacts**. The aggregate LCR increases associated with warming mask considerable differences among the individual causes. On a percent basis, losses associated with drought and heat are projected to increase by 92 and 105%, respectively, for corn and by 59 and 105%, respectively, for soybeans ($p$ value < 0.01). Conversely, losses associated with excess moisture and cold are projected to decrease by 43 and 256%, respectively, for corn and by 59 and 234%, respectively, for soybeans ($p$ value less than 0.05). However, in absolute terms the combined drought and heat increases are significantly larger than the combined excess moisture and cold decreases, thereby leading to a net increase in overall losses. As with the aggregate LCR, we find evidence of widespread spatial heterogeneity of the cause-specific losses associated with warming (Fig. 5).

## Discussion

This research leverages approximately 30,000 observations across >1600 counties from 1989 to 2014 for both corn and soybeans to show that moderate warming of 1 °C is associated with increased production risk and increased occurrence of both drought and heat losses. Production risk has important implications for on-farm decision-making and food supply. At the farm level, producers make annual decisions on input expenditures (e.g., seed, fertilizer) based on the distribution of potential outcomes (e.g., yield, revenue) they face. When those outcomes are riskier, producers may devote fewer inputs overall into the production process or increase their use of risk-reducing inputs in the same way that investors shy away from risky stocks[57,58]. In addition, extreme heat and drought occurrences tend to be spatially widespread, thereby correlating production shocks across farms.

Thus, in the aggregate, we expect warming to both decrease food supply on average and increase intraseasonal variability.

A major strength of these data is the attribution of losses to >40 different causes. Here we focus on heat, cold, drought, and excess moisture to demonstrate the nuanced effects of warming temperatures, which not only increase losses associated with drought and heat but also decrease losses associated with cold and excess moisture. This is perhaps unsurprising, but our approach also quantifies the relative changes for each cause, which can directly inform adaptation needs at both the local and aggregate levels. We also find that the increase in production losses for drought are much larger than those for heat and that the combined heat/drought increases are larger than the combined excess moisture/cold decreases on average across all counties in the sample. However, there is substantial spatial heterogeneity of these effects. Adaptation will therefore not look the same everywhere, even within the U.S. corn belt, a large globally influential production region that is often considered homogeneous in its production practices. Thus there will not be a uniform silver bullet solution for policy supporting climate change adaptation through technology innovation but rather a wide range of new technologies tailored to localized needs.

Separate from potential policies focusing on adaptation, our findings suggest that warming temperatures, by increasing production risk, can have large effects on the program costs of existing government-supported risk management policies for agriculture. The U.S. FCIP has become the cornerstone of public support for U.S. agricultural producers, with over $110 billion in liabilities in 2018 (RMA 2019). It has influenced the creation and design of similar programs globally in both developed and

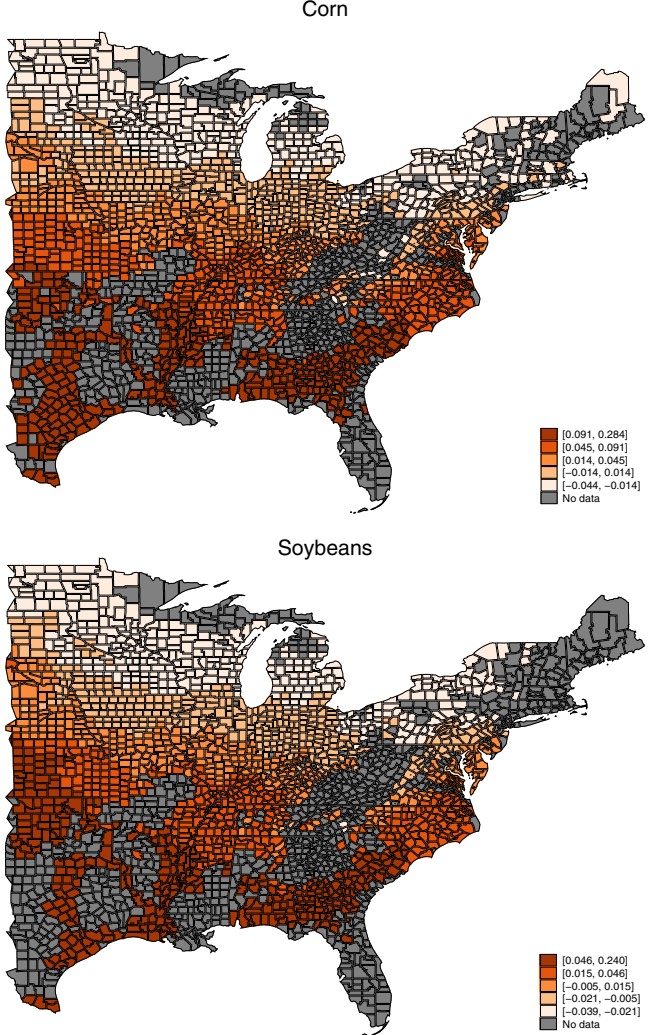

**Fig. 4 Spatial distribution of a 1 °C warming impact on the county-level average aggregate loss–cost ratio (LCR), 1989–2014 ($N = 30{,}261$ for corn and $N = 29{,}014$ for soybeans).** For the uniform 1 °C warming scenario, the predictions are based on the estimated coefficients of the total LCR model (assuming no precipitation changes). The darker colors represent greater impacts on the average LCRs.

developing countries, with India and China comprising the next two largest agricultural insurance markets. These programs are typically heavily subsidized by the government, which integrates program effects into the general economy via taxpayer burden. It also means that increased production risk will increase the out-of-pocket cost of purchasing insurance for producers, which may in turn increase the governmental outlay needed to subsidize this purchase. In addition to the implication of higher program costs, our cause-specific analyses can provide insights on the feasibility and costs of new insurance product developments. To give an example, developing new products (e.g., single peril or index insurance products), which target specific causes that are relatively more sensitive to climate change or difficult to adapt to, may reduce the cost of operating crop insurance programs.

It is worth noting some of the limitations of our work, as well as directions for future research. One unique feature of the COL data is that losses are reported on a monthly basis. Thus the analysis presented here could be modified such that monthly losses are matched to monthly weather variables. This could provide unique insights into the timing of intraseason warming

impacts on the different causes of yield losses. There are also a number of assumptions concerning the empirical approach that could be relaxed and/or modified. Some of these include: assuming a different growing season, allowing for weather impacts to differ at different times during the growing season, and allowing for weather impacts to differ across space and/or time. Farmer adaptation could potentially be incorporated by endogenizing the growing season or by building a more structural model that features certain adjustments in the crop insurance program not explicitly captured by our chosen approach. The approach taken here also assumed unchanged precipitation and $CO_2$ levels in response to warming. Permitting these factors to change could potentially provide additional insights into how warming impacts yield losses.

## Methods

**Data sources**. The analysis in this paper uses data from several different sources. The crop insurance data were obtained from the SOB and COL databases, both publicly available datasets maintained by the RMA. The SOB data contains county–year–specific observations of total losses and liabilities by crop. The COL data contains crop–county–year-specific observations of losses disaggregated by each of the 44 different possible causes. We restrict our analysis to corn and soybeans from 1989 to 2014. These are the two most widely grown and insured crops in the United States, and 1989–2014 is the period for which we could obtain the required information on crop insurance. Precipitation and weather data come from the Parameter-elevation Regressions on Independent Slopes Model (PRISM) dataset. Similar to Schlenker and Roberts[6], we compute daily degree days above each 1 °C threshold using the sinusoidal curve method[59] and then aggregate degree days that fall within the assumed growing season of April to September.

When a claims adjuster determines that indemnities be paid, they cite a specific cause associated with the loss. This determination is based on the weather history, the timing of the loss, and other information supplied by the producer. There may be instances in which the chosen cause is not the true cause or that a particular loss was the result of multiple causes. This type of issue can be described as measurement error in the dependent variable, which will not bias the estimated results. Moreover, even if the assigned cause is not always correct, it is the result of the subjective interpretations of the adjusters and producers that will guide future producer behavior and adaptation. For example, if a producer perceives yield losses to be more the result of drought, rather than heat stress, this will likely incentivize different adaptation behavior (e.g., irrigation).

In any given year, there are in excess of 30 possible causes, many of which are seldom invoked. Figure 1 presents the distribution of indemnities by cause for corn and soybeans during the 1989–2014 period. For both crops, the largest cause of insurance payments is drought, which accounted for about 45% of payments in corn and 39% in soybeans. Excess precipitation also played an out-sized role, particularly in soybeans.

**Empirical framework**. We use a framework similar to the empirical specification in Schlenker and Roberts[6]. Specifically, let $i$ denote a county and $t$ denote a year. We estimate regressions of the following form:

$$
\begin{aligned}
y_{it} = c_i + f(\tau_{it}, \beta) &+ \gamma_1 \text{Prec}_{it} + \gamma_2 \text{Prec}_{it}^2 + \sum_s \delta_{1s} D_s \times \text{Time}_t \\
&+ \sum_s \delta_{2s} D_s \times \text{Time}_t^2 + u_t + \varepsilon_{it},
\end{aligned}
\tag{1}
$$

where $y_{it}$ is the LCR, defined as $\text{LCR}_{it} = \left( \sum_j \text{Losses}_{it}^j \right) \left( \text{Liabilities}_{it} \right)^{-1}$. $\text{Losses}_{it}^j$ is the dollar value of realized indemnity payments in county $i$ in year $t$ due to cause $j$, and $\text{Liabilities}_{it}$ is the maximum dollar value of indemnity payments in county $i$ in year $t$. The predictor variables are defined as follows: $\text{Prec}_{it}$ denotes total precipitation, $D_s$ is an indicator variable for state $s$, $\text{Time}_t$ is a time trend variable, and $c_i$ and $u_t$ are county fixed effects and year fixed effects, respectively. The summation terms are state-specific quadratic time trends. The function $f(\tau_{it}, \beta)$ relates county- and time-specific temperatures (measured in degree days following Snyder[59]), denoted by $\tau_{it}$, to $y_{it}$. Similar to Schlenker and Roberts[6], our specification for $f(\tau_{it}, \beta)$ is a piecewise linear function: we permit the impact of temperatures on $y_{it}$ to vary over three different intervals in a piecewise linear fashion.

In a separate set of regressions, we estimate temperature and precipitation impacts on the cause-specific LCRs. Specifically, we can write $y_{it} = \sum_j y_{jit}$, where $j$ denotes a cause that could be cited by the claims adjuster (e.g., drought). We focus our attention on the main weather-related causes, particularly those that relate to temperature and moisture. These include heat, cold, drought, and excess moisture.

The piecewise linear function contains cutoff points that need to be determined during estimation. To obtain the cutoffs, we estimate Eq. (1) with different cutoff candidate values and then select the cutoffs that best fit the data. We place certain restrictions on the candidate values. Specifically, we fix the first cutoff at 10 °C and then search for two cutoffs, one between 20 °C and 35 °C and the other

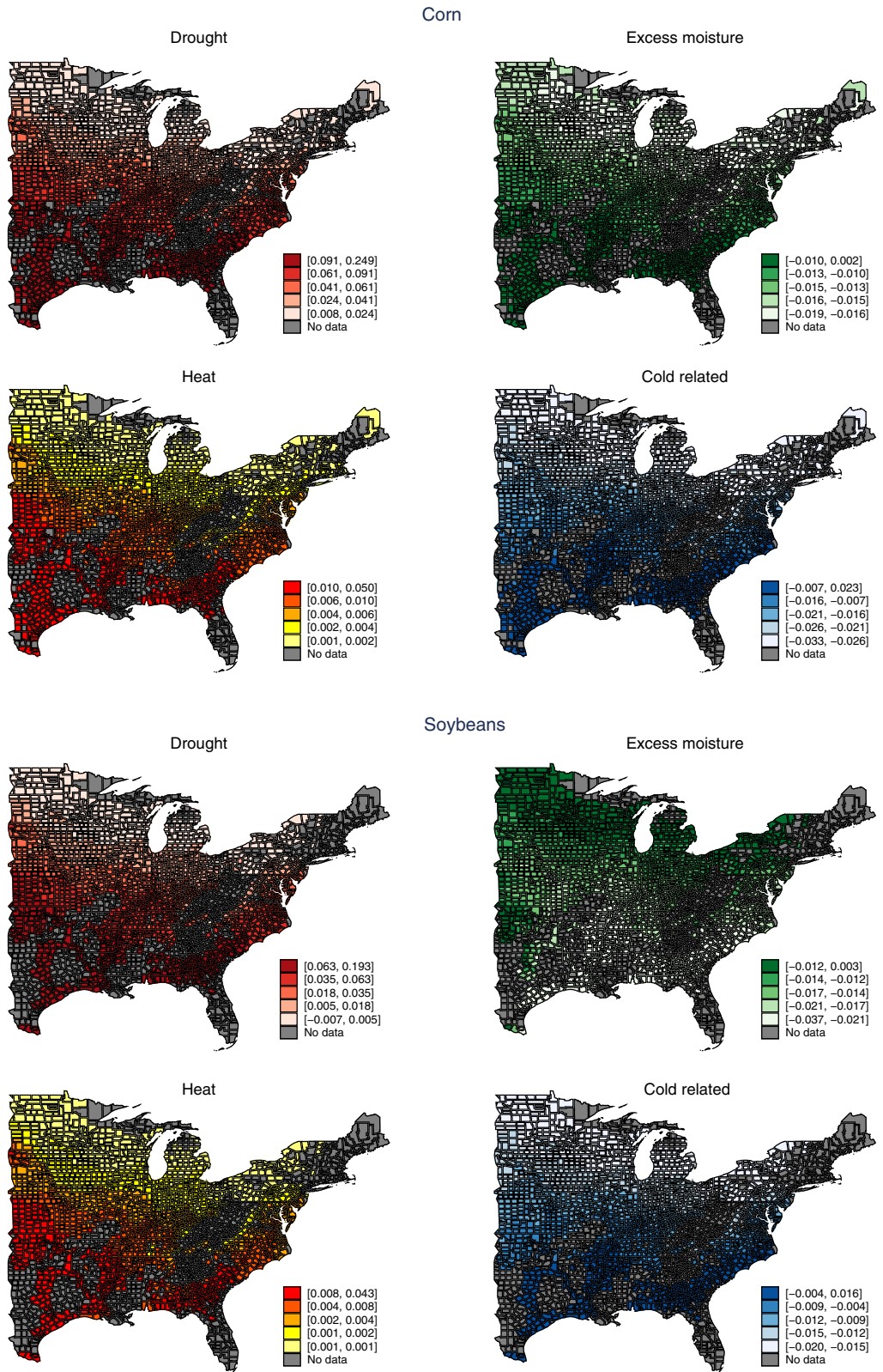

**Fig. 5 Spatial distribution of 1 °C warming impact on the county average cause-specific loss–cost ratio (LCR), 1989–2014 ($N = 30,261$ for corn and $N = 29,014$ for soybeans).** For the uniform 1 °C warming scenario, the predictions are based on the estimated coefficients of the four cause-specific LCR models (assuming no precipitation changes). The darker colors represent greater impacts on the average LCRs.

between 36 °C and 43 °C. Supplementary Table 1 presents the goodness-of-fit measure for our five key dependent variables—the overall LCR and the four cause-specific LCRs—and the corresponding two temperature cutoffs. For the overall LCRs, the first cutoffs are 27 °C for corn and 28 °C for soybeans. The cutoffs for heat, drought, and excess moisture losses are similar but a bit higher, and the cutoffs for cold-related losses are lower at 24 °C and 36 °C. The warming effects are obtained by estimating $f(\bar{\tau}_{i1}, \hat{\beta}) - f(\bar{\tau}_{i0}, \hat{\beta})$, where $\bar{\tau}_{i1}$ is the vector of the averages of temperature variables under 1 °C warming scenario, $\bar{\tau}_{i0}$ is the vector of the average of the observed temperature variables, and $\hat{\beta}$ is the vector of estimated coefficients for the piecewise linear function of temperature effects. We cluster standard errors by year for our main specification. All regression analyses were conducted using the statistical software Stata/SE 15.

**Reporting summary**. Further information on research design is available in the Nature Research Reporting Summary linked to this article.

## Data availability
The Cause of Loss and Summary of Business datasets can be obtained at https://www.rma.usda.gov/SummaryOfBusiness/CauseOfLoss. The PRISM weather data can be obtained at http://www.columbia.edu/ws2162/links.html and http://prism.oregonstate.edu. Further details concerning preparation of the data are available at https://github.com/jisangyu-agecon/LCR_Perry_et_al.git.

## Code availability
All code used for this study is available at https://github.com/jisangyu-agecon/LCR_Perry_et_al.git.

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

## Author contributions

E.D.P. and J.Y. conceived the project. E.D.P and J.Y. prepared the data. All the authors analyzed the data, interpreted the results and advised on presentation of the main findings, and contributed to the writing of the paper.

## Competing interests

The authors declare no competing interests.
