## [Peer Review File · Nature Communications]

Reviewers' comments:

Reviewer #1 (Remarks to the Author):

In this paper the authors analyze county-level dataset of crop insurance outcomes, linking paid insurance claims for corn and soybean yields to growing season weather. They find substantial increases in heat and drought losses with higher temperatures, and declines in losses due to excess moisture and cold. Findings are robust to a variety of alternate specifications. While use of the RMA data is novel, and the statistical methods are sound, findings largely confirm already-established facts about how rainfed US corn and soybean yields respond to temperature and rainfall fluctuations. Therefore I do not believe the significance of the study is sufficient to justify publication in a high-profile, general science journal such as Nature Communications.

The main assertion of novelty in the paper seems to come from using the cause of loss information to attribute insurance claims to heat, drought, extreme cold, or excess precipitation. I am convinced neither that the authors are able to do this, nor that we learn more from this approach than from a more direct estimate based on yields. Firstly previous work has already established that much of the adverse yield effects from hot temperatures occur via lowering soil moisture through more rapid evapotranspiration (Lobell et al. 2013; Ortiz-Bobea et al. 2019). Moreover, the agricultural definition of drought includes both heat and precipitation declines. Therefore the theoretical and statistical distinction between yield losses attributed to heat and yield losses attribute to drought is unclear. Secondly, what exactly is captured in the assessor's attribution to these different causes, and why this (as opposed to yield losses themselves) should be an interesting object of study is also not clear. If the question is whether extreme cold or excess moisture or precipitation deficits drives yield losses, that question could be answered more directly by regressing yield changes on the relevant measures of growing season weather (i.e. extreme cold and peak rainfall intensity).

I am also concerned that the weather variables used, as well as the climate change projections, are not sufficiently sophisticated to answer the question the authors are interested in. If the aim of the paper is to model the growing-season weather predictive of yield losses attributed to different causes, I think more thought should be given to appropriate weather variables. For instance, explaining losses due to excess moisture using only a quadratic in growing season precipitation seems inappropriate. Standard measures of rainfall intensity include variables like maximum cumulative rainfall over a 5-day period and should be considered. Similarly, investigating interaction effects between extreme hot temperatures and rainfall for explaining drought and heat outcomes would also be interesting.

For their climate change projections, the authors use a 1 degree uniform warming. Using a projection derived from climate models that accounts for geographic heterogeneity in the rate of warming would be preferred, though probably would not change results substantially given the relatively small geographic area of the study. More concerning for me is their statements about the effect of climate change on extreme precipitation losses that only consider the effects of warmer temperatures. It is well-established that climate change will lead to increases in rainfall intensity in many areas. Claiming that climate change will decrease yield losses due to extreme rainfall without properly accounting for this effect is a major problem.

My final concern is more minor, but relates to the interpretation of both the adaptation tests and the climate change projections. My understanding is that yield losses are defined relative to a rolling historical baseline. Therefore, if yields gradually change because of climate change, is it necessarily true that this will lead to an increase in insured losses in the manner calculated by the authors? I think this

issue complicates the interpretation both of the adaptation regressions, particularly those using the rolling mean, and the climate change projections and deserves more discussion.

Reviewer #2 (Remarks to the Author):

This was an interesting paper that exploits the RMA loss data. Although the results aren't surprising given the vast literature on yield effects and the close ties from yield to claims, it is still useful in that it looks at all perils at once and can examine net effects. Overall I'm in favor of publishing, with a few pending issues.

-the authors look at April-september, which makes sense for the heat and drought indices. But I'm wondering if the window should be wider to look at frost and other losses. Isn't late fall frost a big issue in some places? Hail certainly seems important in fig 1, even more so than cold wet weather.

-Similarly, they seemingly only look at temperatures above 10c, but I would think hail and freeze would respond to lower temperatures. By leaving these out they open themselves up to criticism that they are ignoring some of the benefits and thus overstating net losses. I suggest more attention to these specific causes. Relatedly, it's not clear what 'cold related' causes refers to. Fig 1 does not include that term. Are they lumping various causes together?

In both abstract and discussion, the authors note that drought losses far exceed heat losses. I think it's important to say this is based on reported causes, rather than the actual underlying causes. As the authors note, a lot of drought stress is caused by heat and they could be clearer about this throughout in terms of language.

Fig 4 & 5 units are non-intuitive? Maybe better to express as % change rather than raw LCR change. This will make the figure more widely readable and used.

I think fig 2 and 3 are probably not as useful to the story as figures A5&6. I also think A5&6 could be combined into a single figure for the main paper.

Start of results should be moved to end of intro

REVIEWER RESPONSES FOR “USING INSURANCE DATA TO QUANTIFY THE IMPACT OF CLIMATE ON YIELD RISK AND ITS CAUSES”

Revision Summary

1. Responses to each reviewer’s comments are in bold. We have also highlighted all changes in the manuscript.
2. We restructured parts of the manuscript and added new discussion that highlights and clarifies the novelty and contribution of our study. In particular, the new Section A3 and table A6 of the Supplementary Appendix (SA) demonstrate the advantage of using loss data instead of yield data to quantify the impact of warming on yield risk. Related discussion has been added to p.4 of the manuscript.
3. Following both reviewers’ comments, we have added new analyses and robustness checks to justify our approach. In particular, table A2 provides new warming impacts with a wider growing season (March-October) and with an additional cold temperature variable included in the model. We have also adjusted our language in the abstract and paper concerning the distinction between reported and actual causes, especially as it concerns drought and heat losses.
4. Following Reviewer #2’s suggestion, we moved what was formerly Figure 3 to the SA (now Figure A5), and what were formerly Figures A5 and A6 to the manuscript (now Figure 3).

Reviewer #1 (Remarks to the Author):

In this paper the authors analyze county-level dataset of crop insurance outcomes, linking paid insurance claims for corn and soybean yields to growing season weather. They find substantial increases in heat and drought losses with higher temperatures, and declines in losses due to excess moisture and cold. Findings are robust to a variety of alternate specifications. While use of the RMA data is novel, and the statistical methods are sound, findings largely confirm already-established facts about how rainfed US corn and soybean yields respond to temperature and rainfall fluctuations. Therefore I do not believe the significance of the study is sufficient to justify publication in a high-profile, general science journal such as Nature Communications.

The main assertion of novelty in the paper seems to come from using the cause of loss information to attribute insurance claims to heat, drought, extreme cold, or excess precipitation. I am convinced neither that the authors are able to do this, nor that we learn more from this approach than from a more direct estimate based on yields.

While we agree that one of the main novelties of our research is to use the cause of loss information to better understand how temperature variation impacts yield losses, we respectfully disagree that this is the only, or even primary, contribution of our study. One of the main benefits of using the loss data is that they can inform on how temperature impacts *yield risk*, as contrasted with mean yields. By yield risk, sometimes referred to as yield sensitivity (see, e.g., Lobell et al. (2014) and Ortiz-Bobea, Knippenberg, and Chambers

(2018), we mean the propensity of farm-level yields to fall below expected yields (or certain portions of expected yields). It is entirely possible, for example, that higher temperatures reduce mean yields, but that yield risk remains unchanged or even decreases. We now highlight this distinction on pp. 2-3 of the manuscript.

The distinction between mean yield and yield risk is similar to the distinction between average returns and downside measures such as Value-at-Risk (VaR) or Conditional Value-at-Risk (CVaR) in the finance literature (e.g. Engle, and Manganelli 2004; Kuester, Mittnik, and Paolella 2006). Estimating downside risk measures such as VaR or CVaR requires more rigorous and complex approaches than estimating (conditional) mean equations. With respect to yield risk, one *could* use yield information alone to assess how temperature variation impacts yield risk, and this has indeed been done, but there are some important limitations to using yield data. Chiefly, yield data is generally only available at the county level as an average of farm-level yields. Insofar as farm-level yields are not perfectly correlated within a county, analyses that use this type of data will tend to under-estimate the impact of temperature on yield sensitivity.

To demonstrate this aggregation problem, we have added a new simulation exercise to the Supplementary Appendix (SA). In the simulation, we show that using yield data only leads to predictions that underestimate the impact of temperature on yield sensitivity. By contrast, using loss data correctly predicts the impact of a one-degree increase on losses. Overall, we believe that directly estimating how weather variables affect the loss-cost ratio gives us useful and unique insights into the dynamics of downside risk in food production, and this is because the loss-cost ratio is a good proxy for such risk. Details and discussion regarding the simulation can be found in section A3 and table A6 of the SA. Within in the manuscript, related discussion has been added to the bottom of p. 4.

Firstly previous work has already established that much of the adverse yield effects from hot temperatures occur via lowering soil moisture through more rapid evapotranspiration (Lobell et al. 2013; Ortiz-Bobea et al. 2019). Moreover, the agricultural definition of drought includes both heat and precipitation declines. Therefore the theoretical and statistical distinction between yield losses attributed to heat and yield losses attribute to drought is unclear. Secondly, what exactly is captured in the assessor's attribution to these different causes, and why this (as opposed to yield losses themselves) should be an interesting object of study is also not clear.

We agree that there is an important distinction between the causes cited by adjusters and the true causes of yield losses. However, we disagree that the former is not an important source of information. To the contrary, from an adaptation perspective, understanding what producers believe to be the cause of losses may be as important as the true underlying causes. To give an example, if farmers primarily perceive losses to have been caused by drought, rather than heat stress, they are probably more likely to use certain forms of adaptation, such as irrigation. By contrast, if they perceive losses to be primarily caused by heat stress, this may incentivize other forms of adaptation, such as changing which varieties they purchase.

Having said this, in the abstract we are now more explicit by noting that these are reported causes:

“By using cause of loss information, we also find that additional losses under warming primarily result from additional reported occurrences of drought, with reported losses due to heat stress playing a much smaller role, and some types of predicted losses such as excess precipitation and cold related damages actually decreasing under warmer temperatures. There is also significant spatial heterogeneity in the predicted impacts of warming.”

We have also included additional discussion on the distinction between the causes cited by adjusters and the true causes of yield losses. On p. 6 of the manuscript, we write:

“It is important to recognize, however, that the determination of what causes a particular loss is based on the interpretations of adjusters and producers, which may not always align exactly with the "true" cause of losses. Nonetheless, the perceptions of adjusters and producers are just as important for understanding the ramifications of climate change, as they likely reflect how farmers adjust their behavior in response to warmer temperatures.”

In the original submission we also discussed this issue in the Data Sources subsection of the Methods. We have maintained this material in the current manuscript. On p. 18, we write:

“When a claims adjuster determines that indemnities be paid, they cite a specific cause associated with the loss. This determination is based on the weather history, the timing of the loss, and other information supplied by the producer. There may be instances in which the chosen cause is not the true cause, or that a particular loss was the result of multiple causes. This type of issue can be described as measurement error in the dependent variable, which will not bias the estimated results. Moreover, even if the assigned cause is not always correct, it is the result of the subjective interpretations of the adjusters and producers, which will guide future producer behavior and adaptation. For example, if a producer perceives yield losses to be more the result of drought, rather than heat stress, this will likely incentivize different adaptation behavior (e.g., irrigation).”

If the question is whether extreme cold or excess moisture or precipitation deficits drives yield losses, that question could be answered more directly by regressing yield changes on the relevant measures of growing season weather (i.e. extreme cold and peak rainfall intensity).

Our goal with this aspect of the research was not to determine whether excess cold and moisture drives yield losses---we take this as given. Rather, we wanted to assess how these losses are affected by *changes in temperature*. Linking temperature with specific causes of yield losses is useful because it can tell us which types of adaptation methods are more likely to occur in warming climates.

I am also concerned that the weather variables used, as well as the climate change projections, are not sufficiently sophisticated to answer the question the authors are interested in. If the aim of the paper is to model the growing-season weather predictive of yield losses attributed to different causes, I think more thought should be given to appropriate weather variables. For instance,

explaining losses due to excess moisture using only a quadratic in growing season precipitation seems inappropriate. Standard measures of rainfall intensity include variables like maximum cumulative rainfall over a 5-day period and should be considered. Similarly, investigating interaction effects between extreme hot temperatures and rainfall for explaining drought and heat outcomes would also be interesting.

Thank you very much for pointing out the concern regarding the use of quadratic precipitation variables. We do understand that precipitation is an imperfect proxy for soil moisture. Therefore, as an alternative, we estimated a model that includes Vapor Pressure Deficit (VPD) as an additional control, the results of which can be found in table A4. Overall, we do not find major differences in the predictive power of the model nor in the warming impact estimation. We also want to stress our previous response that our primary focus in this manuscript is to investigate the link between temperature and the various causes of yield losses.

For their climate change projections, the authors use a 1 degree uniform warming. Using a projection derived from climate models that accounts for geographic heterogeneity in the rate of warming would be preferred, though probably would not change results substantially given the relatively small geographic area of the study.

More concerning for me is their statements about the effect of climate change on extreme precipitation losses that only consider the effects of warmer temperatures. It is well established that climate change will lead to increases in rainfall intensity in many areas. Claiming that climate change will decrease yield losses due to extreme rainfall without properly accounting for this effect is a major problem.

We agree that changes in rainfall induced by warming are a potentially important factor to consider. In the manuscript, we discuss this briefly on p. 18:

“The approach taken here also assumed unchanged precipitation and CO2 levels in response to warming. Permitting these factors to change could potentially provide additional insights into how warming impacts yield losses.”

We should also note that one of the main reasons we simulated a warming scenario is because the temperature impacts are difficult to interpret on their own. In other words, the goal of conducting the analyses using the uniform warming scenario is to have a sharper interpretation of the estimated temperature effects. By contrast, the marginal impact of additional precipitation can be more easily assessed from the estimates provided in Figure A7. Unlike the degree day variables, the marginal impact of precipitation does not require re-constructing the variables; rather, it can be computed simply by taking the derivative of the quadratic precipitation equation. Nonetheless, in Figure A7 we now also provide the impact of a 10% increase in precipitation from its mean on each of the cause-specific causes. These impacts suggest that, even with additional precipitation, a warming climate would lead to increases in the overall, drought, and heat LCRs, and decreases in the cold and moisture related LCRs.

My final concern is more minor, but relates to the interpretation of both the adaptation tests and the climate change projections. My understanding is that yield losses are defined relative to a rolling historical baseline. Therefore, if yields gradually change because of climate change, is it necessarily true that this will lead to an increase in insured losses in the manner calculated by the authors? I think this issue complicates the interpretation both of the adaptation regressions, particularly those using the rolling mean, and the climate change projections and deserves more discussion.

This is an excellent point and it is precisely why we included results from models that allow for various forms of adaptation. At their core, the adaptation models consist of the same regressions as the baseline models, but with the weather and loss variables computed over longer time intervals. As discussed in Hsiang (2016), these estimated impacts are therefore inclusive of adaptation, but this comes at the cost of reduced control of unobserved heterogeneity (e.g., through various fixed effects). In our context, this means that these models can capture the fact that a rolling historical baseline would change under a warming climate. Consider, for example, the cross-section models. Here, the estimated parameters are based on whether counties that are warmer, on average, have higher LCRs. The results we provide in Table A5 suggest that, even with adaptation being allowed for, a warming climate would lead to increases in the overall LCR of a similar magnitude, albeit slightly smaller, to our baseline estimates.

In the new manuscript, we have added related discussion to p. 13:

“Thus, we conduct an additional set of robustness checks with respect to the consideration of potential adaptation using an approach suggested by Hsiang (2016). We compute warming impacts based on four alternative empirical models: (i) a long-difference model, (ii) a cross section model, and (iii)-(iv) models with five and ten year moving averages of the model variables (SA Table A5). As noted in Hsiang (2016), estimated impacts based on approaches that exploit climatic variations over longer temporal frequencies implicitly allow for changes in beliefs or adaptation by individuals (in this case, farmers and the FCIP).”

Reviewer #2 (Remarks to the Author):

This was an interesting paper that exploits the RMA loss data. Although the results aren't surprising given the vast literature on yield effects and the close ties from yield to claims, it is still useful in that it looks at all perils at once and can examine net effects. Overall I'm in favor of publishing, with a few pending issues.

-the authors look at April-September, which makes sense for the heat and drought indices. But I'm wondering if the window should be wider to look at frost and other losses. Isn't late fall frost a big issue in some places? Hail certainly seems important in fig 1, even more so than cold wet weather.

This is a good point and we agree that cold-related losses are more likely to occur at the beginning and end of the growing season. To address this, we re-estimated the regressions using a March-October growing season. We then simulated the same 1 degree warming

scenario. The results for this wider growing season are presented in Table A2 of the SA, both with and without the inclusion of cold-temperature variables. Overall, the warming impacts are very similar to the baseline impacts. We have therefore elected to maintain the original baseline as the baseline going forward. Related discussion can be found in the first paragraph of section A2 of the SA.

-Similarly, they seemingly only look at temperatures above 10c, but I would think hail and freeze would respond to lower temperatures. By leaving these out they open themselves up to criticism that they are ignoring some of the benefits and thus overstating net losses. I suggest more attention to these specific causes. Relatedly, it's not clear what 'cold related' causes refers to. Fig 1 does not include that term. Are they lumping various causes together?

Your interpretation of our baseline estimates is correct: we implicitly assumed that temperatures under 10c have no marginal impact on the LCR. This seemed a fair assumption for the drought and heat causes, but we agree this may not be true for some of the other causes. As noted in the previous response, in the revised version we have estimated models that include an additional variable for the total number of degree days below 10c. We find that additional days below 10c do increase the overall and freeze LCRs, as expected, but the effects are not statistically significant. The warming impacts associated with models that include a cold temperature threshold can be found in table A2.

Note also that the cause “cold-related” is the sum of cold wet weather, freeze, and frost losses. We note this by adding a clarifying sentence on p. 6 of the manuscript.

In both abstract and discussion, the authors note that drought losses far exceed heat losses. I think it's important to say this is based on reported causes, rather than the actual underlying causes. As the authors note, a lot of drought stress is caused by heat and they could be clearer about this throughout in terms of language.

This is an excellent point. In the original paper we do discuss this issue a bit in the Data Sources subsection of the Methods. For example, on p. 18 of the original manuscript, we wrote:

“When a claims adjuster determines that indemnities be paid, they cite a specific cause associated with the loss. This determination is based on the weather history, the timing of the loss, and other information supplied by the producer. There may be instances in which the chosen cause is not the true cause, or that a particular loss was the result of multiple causes. This type of issue can be described as measurement error in the dependent variable, which will not bias the estimated results. Moreover, even if the assigned cause is not always correct, it is the result of the subjective interpretations of the adjusters and producers, which will guide future producer behavior and adaptation. For example, if a producer perceives yield losses to be more the result of drought, rather than heat stress, this will likely incentivize different adaptation behavior (e.g., irrigation).”

In the abstract, we are now more explicit by noting that these are reported causes:

“By using cause of loss information, we also find that additional losses under warming primarily result from additional reported occurrences of drought, with reported losses due to heat stress playing a much smaller role, and some types of predicted losses such as excess precipitation and cold related damages actually decreasing under warmer temperatures. There is also significant spatial heterogeneity in the predicted impacts of warming.”

We have also added the following sentence to p. 6 of the manuscript to clarify that the subjective assessments of producers and claims adjusters may not reflect the true underlying causes of yield damages:

“It is important to recognize, however, that the determination of what causes a particular loss is based on the interpretations of adjusters and producers, which may not always align exactly with the "true" cause of losses. Nonetheless, the perceptions of adjusters and producers are just as important for understanding the ramifications of climate change, as they likely reflect how farmers adjust their behavior in response to warmer temperatures.”

Fig 4 & 5 units are non-intuitive? Maybe better to express as % change rather than raw LCR change. This will make the figure more widely readable and used.

This is a good point. Because the LCR is, in principle, bounded between 0 and 1, the estimated impacts are, in fact, percentage point impacts. As depicted in Figure A3, in some cases the LCR can exceed 1 which can result from sampling errors or accounting issues. In any case, to make this clearer, we have added the following sentence to p. 10 the manuscript:

“These graphs depict the impact of one additional day at each temperature on the overall LCR. For example, one additional day at 27 C instead of a day below 10 C within the fixed growing season reduces the LCR by a bit under 0.01 or one percentage point (or, put differently, by one cent per dollar of liability).”

I think fig 2 and 3 are probably not as useful to the story as figures A5&6. I also think A5&6 could be combined into a single figure for the main paper.

After some reflection, we agree with this sentiment and have therefore moved what was formerly Figure 3 to the appendix (now Figure A5), and what were formerly Figures A5 and A6 to the manuscript (now Figure 3).

Start of results should be moved to end of intro

The first paragraph of the Results section was moved to the end of the Introduction.

References

Engle, R.F. and Manganelli, S., 2004. CAViaR: Conditional autoregressive value at risk by regression quantiles. *Journal of Business & Economic Statistics*, 22(4), pp.367-381.

Kuester, K., Mittnik, S. and Paolella, M.S., 2006. Value-at-risk prediction: A comparison of alternative strategies. *Journal of Financial Econometrics*, 4(1), pp.53-89.

Hsiang, S. Climate econometrics. *Annual Review of Resource Economics* 8, 43–75 (2016).

Lobell, D.B., Roberts, M.J., Schlenker, W., Braun, N., Little, B.B., Rejesus, R.M. and Hammer, G.L., 2014. Greater sensitivity to drought accompanies maize yield increase in the US Midwest. *Science*, 344(6183), pp.516-519.

Ortiz-Bobea, A., Knippenberg, E. and Chambers, R.G., 2018. Growing climatic sensitivity of US agriculture linked to technological change and regional specialization. *Science advances*, 4(12), eaat4343.

REVIEWERS' COMMENTS:

Reviewer #1 (Remarks to the Author):

In my original comments I stated that I was not convinced that the novelty or importance of these results were sufficient to justify publication in Nature Communications. The authors argue in their response two things: 1) that the LCR captures an important measure of downside yield risk that cannot be recovered in regressions using county mean yields and 2) using reported cause of loss is important in understanding adaptations.

I do agree that in absence of farm-level yields, this measure captures something about the intra-county distribution of negative yield shocks. I am unconvinced that showing this measure changes in similar ways to what has already been well established for mean yields constitutes a substantive advance. The other argument that perceived cause of loss is important in understanding adaptations is also unpersuasive: The example given for the importance of distinguishing between perceived drought vs heat losses is irrigation, but irrigation is protective against both heat and drought. Moreover, if an important contribution of the paper is assessing what types of adaptations are likely to occur in a warming climate, as stated in the rebuttal, the paper does not do that. Based on the rebuttal and the revised manuscript, I do not see a reason to change my initial opinion that the originality and novelty of the contribution is not sufficient to justify publication in a high-profile, interdisciplinary science journal. But that is ultimately an editorial decision.

I do appreciate the trouble the authors have gone to to control for additional weather variables, particularly cold days and VPD. The robustness of findings to these additions is reassuring.

Specific Comments

- The authors only consider a uniform warming scenario, not changes in rainfall or CO₂, and so the title should be changed to "...Quantify the Impact of Warming on Yield Risk". As I stated in my initial comments, climate change will increase rainfall intensity – an effect that is well understood and has already been detected in rainfall patterns at the global scale. It seems likely this will increase future crop losses from extreme moisture, but this effect is not examined by the authors. Therefore they should be clear that they are only considering the effects of temperature change.
- Similarly, in figure captions for projections (e.g. Figure 5), the assumption "assuming no change in rainfall" should be specified explicitly
- In the simulation, the authors state that a "more realistic scenario" is that temperature affects both mean yields and the within-county, intra-annual variance in yields. It is not at all clear to me why the within-county variance (e.g. driven by local difference in soil quality, rainfall, farmer management etc) should be affected by temperature, or why that effect would obviously be an increase in variance.

Reviewer #2 (Remarks to the Author):

sorry for delay. i think the authors have done a good job responding to the concerns, and it is a nice contribution to the literature.

Reviewer #1 (Remarks to the Author):

In my original comments I stated that I was not convinced that the novelty or importance of these results were sufficient to justify publication in Nature Communications. The authors argue in their response two things: 1) that the LCR captures an important measure of downside yield risk that cannot be recovered in regressions using county mean yields and 2) using reported cause of loss is important in understanding adaptations.

I do agree that in absence of farm-level yields, this measure captures something about the intra-county distribution of negative yield shocks. I am unconvinced that showing this measure changes in similar ways to what has already been well established for mean yields constitutes a substantive advance.

We appreciate that you agree the LCR variable is better able to capture downside yield risk, but we strongly differ on your conclusions. First, the finding in the literature that warming reduces mean yields does not necessitate that warming also increases downside yield risk around the new mean yield levels; these directional impacts need to be measured separately. In addition, even if the results here confirm previous suspicions, the magnitude of the risk effects are important to quantify. Second, changes in the volatility of yields have important and completely separate implications from changes in mean yields. Our study documents one such implication—premium rates will increase---for an insurance market that now exceeds \$100 billion in annual liabilities. These conclusions can be extended to other related domains where the volatility of crop output is of major importance—commodity markets, financial markets, and storage planning, just to name a few. Finally, as we show in the simulation, using county-level yields (which are very common in the crop yield / climate change literature) to assess the impact of warming on downside risk significantly under-estimates the impacts.

The other argument that perceived cause of loss is important in understanding adaptations is also unpersuasive: The example given for the importance of distinguishing between perceived drought vs heat losses is irrigation, but irrigation is protective against both heat and drought. Moreover, if an important contribution of the paper is assessing what types of adaptations are likely to occur in a warming climate, as stated in the rebuttal, the paper does not do that.

We first want to stress that the primary contribution of this paper is to rigorously estimate the impact of warming temperatures on yield risk, with appropriate conclusions and implications drawn (e.g., that the individual cost of federal crop insurance will increase). The cause-specific analysis is an additional contribution and is more speculative in nature, but this is a general feature of this area of study. Indeed, the existing literature widely disagrees on the relative contribution of heat and water stress to yield losses. In our view, therefore, it would be remiss to not analyze and report an untapped and insightful aspect of the RMA insurance data. With that said, we do note at multiple junctures in the manuscript the limitations of the cause-specific data.

Concerning the irrigation example, while it is true that irrigation may be protective against both heat stress and drought, the nature of irrigation measures (timing, intensity, etc.) will differ depending on the perceived source of crop losses. In addition, there are many other examples in which adaptation may differ depending on whether the primary concern is drought or heat. Two such examples include crop variety choice (noted in the manuscript) and the timing of cropping practices (planting, harvesting, etc.). The overall point is that each of these adaptations will differ to some degree depending on whether heat or drought is the main concern.

Specific Comments

The authors only consider a uniform warming scenario, not changes in rainfall or CO₂, and so the title should be changed to "...Quantify the Impact of Warming on Yield Risk". As I stated in my initial comments, climate change will increase rainfall intensity – an effect that is well understood and has already been detected in rainfall patterns at the global scale. It seems likely this will increase future crop losses from extreme moisture, but this effect is not examined by the authors. Therefore they should be clear that they are only considering the effects of temperature change.

This is a valid point, although we do also estimate the impact of precipitation on the LCR. Nonetheless, our emphasis is indeed on temperature, so we have elected to change the title to "Using Insurance Data to Quantify the Multidimensional Impacts of Warming Temperatures on Yield Risk"

Similarly, in figure captions for projections (e.g. Figure 5), the assumption "assuming no change in rainfall" should be specified explicitly

The warming figures now all include the statement "assuming no precipitation changes".

In the simulation, the authors state that a "more realistic scenario" is that temperature affects both mean yields and the within-county, intra-annual variance in yields. It is not at all clear to me why the within-county variance (e.g. driven by local difference in soil quality, rainfall, farmer management etc) should be affected by temperature, or why that effect would obviously be an increase in variance.

Because temperature interacts with the factors you list---soil, rainfall, management, etc.---it will most certainly impact the variance of yields. A reasonable prediction is that that variance would increase but our conclusions would remain unchanged if the variance decreased. In fact, this is precisely the idea behind using the LCR variable to identify whether temperature impacts the higher moments of the yield distribution. In any case, if the variance decreased, using county-level yields to estimate warming impacts would still produce biased impact estimates, whereas using the LCR would not.